# Physical Treatment Reduces Trypsin Inhibitor Activity and Modifies Chemical Composition of Marama Bean (*Tylosema esculentum*)

**DOI:** 10.3390/molecules27144451

**Published:** 2022-07-12

**Authors:** Funmilola Alabi, Elijah G. Kiarie, Caven Mguvane Mnisi, Victor Mlambo

**Affiliations:** 1Department of Animal Science, School of Agricultural Science, North-West University, Mafikeng 2745, South Africa; 23257539@nwu.ac.za; 2Department of Animal Biosciences, University of Guelph, Guelph, ON N1G 2W1, Canada; ekiarie@uoguelph.ca; 3Food Security and Safety Focus Area, Faculty of Natural and Agricultural Science, North-West University, Mafikeng 2745, South Africa; 4School of Agricultural Sciences, Faculty of Agriculture and Natural Sciences, University of Mpumalanga, Nelspruit 1200, South Africa; victor.mlambo@ump.ac.za

**Keywords:** autoclaving, cooking, marama bean, soaking, soybean, trypsin inhibitor activity

## Abstract

The utility of the marama bean (MB) as an alternative protein source to soybean (SB) can be limited by the high concentration of trypsin inhibitors (TI). The physical treatment of MB has the potential to ameliorate the antinutritional activities of TI and modify other chemical components. Thus, this study investigated the effects of physical treatments on the chemical components and trypsin inhibitor activity (TIA) of raw MB and SB. The bean substrates were subjected to each of the following treatment methods: (1) room temperature (20–22 °C) soaking for 24 h; (2) electric stove cooking at 100 °C for 10, 20, and 30 min; (3) steam autoclaving at a temperature of 110 °C and pressure of 7 pounds per square inch (psi), as well as a temperature of 121 °C and 7 psi for 5, 15, and 30 min; (4) pre-soaked autoclaving at 110 °C (7 psi) and 121 °C (17 psi) for 5, 15, and 30 min. Treated MB and SB had greater (*p* < 0.05) crude protein content than untreated samples. All the treatments (except 24 h soaking of MB) reduced (*p* < 0.05) the TIA and ash content. Marama and SB are similar in protein content, but their amino acids profile and TIA are quite different. Soaking for 24 h was less effective in reducing TIA in MB and SB, compared to the thermal methods, and it was detrimental to the ash and amino acids profile of the two beans. Soaking prior to autoclaving yielded beans with the lowest TI concentrations. In conclusion, thermal methods reduced the TI contents and modified the level of proximate components and amino acids profile of the beans.

## 1. Introduction

The marama bean (MB; *Tylosema esculentum*) is an indigenous legume in southern Africa that is directly consumed by indigenous people of Botswana, Namibia, and South Africa [1]. The matured seeds are used for producing porridge, oil, and butter [2]. Marama nutritionally contains oil ranges between 24–48%, predominantly mono- and di-unsaturated fatty acids and without cholesterol [1]. It is also a good source of potassium, phosphorus, calcium, magnesium, sulphur [3], iron, zinc, and B vitamins, including folate [1].

While common plant protein sources, such as chickpea, cowpea, and canola meal, have been assessed for their suitability as SB alternatives in animal nutrition [4,5], the utility of MB for this purpose remains unknown. This indigenous legume contains 34–37% crude protein on a dry matter (DM) basis [3,6], which is comparable to that of soybean, which is reported to range from 33 to 48% DM [7]. However, MB contains anti-nutritional factors such as anti-elastase, TI [8] tannins, and phytates [1], similar to other legume grains. Trypsin inhibitor is the primary anti-nutritional factor in raw soybeans [9]. It is a globulin-type protein with a molecular weight of 24,000 dalton and isoelectric point of 4.5 [10]. The mechanism of action differs for trypsin and chymotrypsin [10]; the trypsin inhibitor binds with trypsinogen to yield an irreversible compound disrupting the formation of an active protease. On the other hand, TI action on chymotrypsin is less pronounced, forming a reversible dissociated compound [11]. Trypsin inhibitors in raw soybeans cause stagnant growth, pancreatic hypertrophy, and hyperplasia in farm animals [12]. Marama bean contains significantly higher levels of TIs than in many other legumes [13]. Indeed, TIs constitutes about 20% of the total marama protein [13], which is 2–4 times higher than in many other legumes [8]. This could limit their utilization as a dietary protein source; thus, strategies to ameliorate the activity of TIs in MB should be investigated to improve its utilization, especially in diets of simple non-ruminants.

Soaking is an easy, low-cost treatment that can reduce the concentration of soluble antinutrients, which are eliminated with the soaking solution [14]. Alternatively, various heat methods, including cooking, microwaving, extrusion, toasting, and roasting, have been reported to be useful in decreasing the concentration and activity of heat-sensitive TIs [14,15,16]. Further, growth performance has been reportedly improved (higher body weight gain and lower feed conversion ratio) in broiler chickens that were fed heat-processed SB-containing diets, compared to those fed a raw soybean-containing diet [17], which could be due to an increase in nutrient availability following TI reduction. Autoclaving and cooking are indicated as the main thermal treatments that reduce the concentration of TIs in protein legumes [15]. Therefore, the objective of this study was to investigate the effect of physical treatments, including soaking, cooking, and autoclaving, on the TIA and chemical composition of MB and SB. The study explored the hypothesis that physical treatments would reduce TIA and improve the nutritive value of MB and SB.

## 2. Materials and Methods

### 2.1. Study Site, Procurement and Processing of the Bean Samples

Whole marama beans harvested in 2019 were obtained from Malwelwe Village, Kweneng District of Botswana (23.94282° S; 25.1999° E), with a soil type referred to as Kalahari sands. The village receives between 350- and 450-mm annual rainfall, with temperatures ranging from 34 to 36 °C during summer and 0 to 5 °C during winter. Whole marama beans were cracked into 2–5 pieces and dehulled individually. Both soybean (hulled) and MB seeds were ground and stored in polythene sample bags under refrigeration pending analyses. Raw soybeans seeds harvested in 2019 were procured from University of Guelph, Feed Mill Unit (Guelph, ON, Canada). Physical treatments and subsequent functional and chemical analyses were conducted at Monogastric Nutrition Laboratory, Department of Animal Biosciences, University of Guelph, Guelph, ON, Canada. Each physical treatment was independently applied to triplicate samples of MB and SB.

**Soaking:** Cracked MB and whole SB samples (20 g per sample) were weighed in triplicate into 250 mL conical flasks and soaked in 200 mL ultra-pure water (Milli-Q IQ 7000, Millipore SAS, Molsheim, France) for 24 h at room temperature (22–25 °C). The samples were then rinsed and drained for about 2 h before oven drying for 48 h at 45–50 °C.

**Cooking:** For each cooking time of 10, 20, and 30 min, 20 g each of SB and MB samples in triplicates were poured into an aluminum pot containing 200 mL (1 g:10 mL) [14] of water on electric coil stove and heated to boiling point (100 °C), after which, cooking time countdown began. Cooked samples were rinsed, drained, and, thereafter, oven-dried for 48 h at 45–50 °C.

**Autoclaving:** Triplicate samples (20 g each) were soaked in ultra-pure water for 24 h at room temperature (22–25 °C), after which soaking water was discarded and samples rinsed twice with fresh water. Both soaked and non-soaked samples were autoclaved for 5, 15, and 30 min at either 110 °C (7 psi) or 121 °C (17 psi) in conical flasks. The ratio of seed to water was 1:7.5 (*w*/*v*) for autoclaving. The heating time countdown began automatically when the internal temperature of the autoclave (3870E Heidolph, Tuttnauer^®^, New York City, NY, United States) reached 110 or 121 °C.

All treated samples were spread in aluminum bowls in one layer and oven dried at 45–50 °C [14] for 48 h, prior to being finely ground, stored in plastic (ziplock) bags, and then refrigerated at 2 °C pending chemical analyses.

### 2.2. Proximate Analysis

The samples of MB and SB, (Table 1) were analyzed by standard methods of Association of Official Analytical Chemists, AOAC, 2005 [18]; dry matter (DM; method 930.15), ash (method 942.05), and nitrogen (method 968.06); nitrogen was assayed by Dumas’ combustion method using Leco Nitrogen analyzer (Leco Corporation, St. Joseph, MI, USA). Crude protein (CP) values were obtained by multiplying assayed Nitrogen values by a factor of 6.25. Gross energy was determined for raw samples by total combustion in a bomb calorimeter (IKA Calorimeter System C 6000; IKA Works, Wilmington, NC, USA). Crude fat analysis was performed via petroleum ether extraction in an ANKOM XT 20 extractor (ANKOM Technology, Fairport, NY, USA). Neutral detergent fiber (NDF) and acid detergent fiber (ADF) of raw samples were assessed according to Van Soest et al., 1991, using Ankom 200 fiber analyzer (ANKOM Technology, Fairport, NY, USA). All determinations were performed in triplicates, and the results were expressed as the mean.

### 2.3. Amino Acids and Trypsin Inhibitor Analysis

Prior to amino acids (AAs) and TI assays, samples of SB and MB were subjected to fat extraction using hexane in the of ratio 1:3 (ground bean: hexane) for 1 h using a magnetic stirrer at low setting. The mixture was allowed to settle for 30 min, and then the hexane decanted. The procedure was repeated three times, with fresh hexane each time. The defatted samples were placed in a fume cupboard overnight to evaporate the n-hexane.

For AAs analyses, samples were digested by acid hydrolysis (AOAC method 982.30). Briefly, SB (50 mg each) and MB (60 mg each) samples were digested in 2.5 mL of concentrated HCl for 24 h at 110 °C, followed by neutralization with 6 N NaOH, and cooled to room temperature. Amino acids in the raw and treated samples were quantified using Norvaline (AccQ-Tag Ultra; Waters Corporation, Milford, MA, USA), as is the amino acid internal standard for ultra-performance liquid chromatography (UPLC; Waters Corporation, Milford, MA, USA).

The TIA of raw and treated samples of defatted MB and SB was determined using spectrophotometric method by American Association of Cereal Chemists (AACC method 22–40) [19], as modified by [20]. This method involves N-benzoyl-DL-arginine p-nitroanilide (BAPA) as a substrate for porcine trypsin, and the ability of aliquots of SB and MB extracts to inhibit the activity of trypsin towards this substrate was utilized to measure the amount of TI in the samples. Briefly, 1 g of finely ground (100-mesh screen) defatted sample of each SB and MB were extracted with 50 mL 0.01 N NaOH/g sample for 3 h with magnetic stirrer at low setting. This was followed by filtration using Whatman filter paper No. 3. The sample extracts were diluted using Tris buffer (0.05 M, pH 8.2; 0.02 M CaCl_2_: 6.05 g hydroxymethylamino methane and 2.94 g CaCl_2_ in 900 mL ultra-pure water, pH was adjusted to 8.2, and the volume brought to 1 L with ultra-pure water) to a point where 1 mL produces trypsin inhibition of 40–60%. Trial dilutions were performed to establish this inhibition value. Portions (0, 0.6, 1.0, 1.4, and 1.8 mL) of diluted extract were pipetted into test tubes and adjusted to 2.0 mL with water. Thereafter, 2 mL of trypsin solution (4 g of porcine trypsin in 200 mL 0.00 1 M HCl) was added to each test tube and placed in a water bath at 37 °C, followed by adding 5 mL substrate solution (40 mg of BAPA hydrochloride was dissolved in 1 mL of dimethyl sulfoxide and diluted to 100 mL with tris-buffer, previously warmed to 37 °C) to the mixture. A freshly prepared BAPA solution was used daily and kept at 37 °C. After 10 min, the reaction was stopped by adding 1 mL of 30% acetic acid solution and mixed using a vortex. The absorbance at 410 nm was measured with a PowerWaveTM XS spectrophotometer (BIO-TEK^®^ Instruments, Inc., Santa Clara, CA, USA) at 23 °C. One trypsin unit is equal to an increase of 0.01 absorbance unit at 410 nm per 10 mL of reaction mixture, in terms of trypsin inhibitor units (TUI).

### 2.4. Statistical Analysis

Proximate composition, amino acids, and TIA data were subjected to a two-way analysis of variance (ANOVA), by means of the general linear model procedure of the Statistical Analysis System (SAS, 2010), with bean substrate and physical treatment methods considered the main factors. Comparison with *p* < 0.05 was considered significantly different, and the means were separated using the probability of difference option in SAS.

## 3. Results

### 3.1. Proximate Composition of Soybean and Marama Bean

The results show that the raw bean samples of the two legumes had comparable dry matter, ash, and crude protein contents (Table 2). However, the crude fat and gross energy content of MB was higher than that of SB. The NDF and ADF content were lower in MB than in SB.

The effect of physical treatments on DM, ash, and CP composition of SB and MB are presented in Table 3. There were significant substrates, treatments, and substrates × treatments interaction effects on all of the chemical components evaluated in SB and MB. The moisture content after oven drying varied across all the treatments and substrates. The values of dry matter of SB were different across all the treatments (*p* < 0.05), with soaking, cooking, and autoclaving resulting in higher values than the raw SB sample. The dry matter values in SB ranged from 90.22 (in raw) to 85.24 %(S2/30). There was no significant difference between cooking for 20 min (C20) and 30 min (C30), with steam autoclaving at 110 °C; 7 psi (A1/30) and 121 °C; 17 psi (A2/30) for 30 min. Similarly, the application of steam autoclaving at 121 °C, 17 psi for 15 min (A2/15), was not significantly different (*p* > 0.05) from presoaked and autoclaved at 121 °C, 17 psi for 5 min (S2/5). Additionally, presoaked and autoclaved at 110 °C, 7 psi for 30 min (S1/30), was not significantly different from presoaked and autoclaved at 121 °C, 17 psi for 30 min (S2/30). The lowest values for cooking, steam autoclaving at 110 °C, 7 psi steam autoclaving at 121 °C, 17 psi presoaked and autoclaved at 110 °C, 7 psi and presoaked and autoclaved at 121 °C, 17 psi were 88.24, 88.05, 86.10, 85.38, and 85.24%, respectively. This shows that, presoaked and autoclaved at 121 °C, 17 psi resulted in the highest loss of 6% in DM content in SB. Dry matter values ranged from 94.03 (in raw) to 87.30% (S2/30) in MB. Similar to SB, all the physical treatments resulted in a DM value lower than the raw MB, and the highest loss of 7% was also caused by presoaked and autoclaved at 121 °C, 17 psi for 30 min (S2/30). The dry matter content in soaked MB was lower than it was at 10 and 20 min of cooking, but higher than the presoaked and autoclaved values. There was no significant difference between 20 min cooking (C20) and steam autoclaving of MB at 110 °C, 7 psi for 5 min (A1/5), 30 min cooking (C30), steam autoclaving of MB at 110 °C, 7 psi for 15 min, presoaked and autoclaved treatments; S2/5 and S2/15. In addition, steam autoclaving of MB; A1/30, A2/15, A2/15, and A2/15 were not significantly different.

The ash content of each SB sample was greater than MB across all treatments (*p* < 0.05). Ash was reduced by soaking, cooking, and autoclaving in SB and MB. Among the treatments, cooking resulted in a smaller reduction in ash content in both SB and MB, compared to the autoclaving methods. Different cooking times (except C10) resulted in similar ash content for each of SB and MB. Autoclaving (S2/5) greatly reduced the ash content of SB by 39% (5.07% in raw sample to the lowest value of 3.07% in S2/5). There was no difference (*p* >0.05) in ash content in cooked (C10, C20 and C30) and steam- and presoaked-autoclaved (A1/5, A1/15, A1/30, A2/5, A2/15, A2/30, S1/5, S1/15, S1/30, S2/15, and S2/30) MB. In addition, the ash content value reduced from 3.06% in raw MB to the lowest of 1.89% (S1/5), leading to highest 38% loss in ash content.

Soybean contained more CP than MB across all the treatments (*p* < 0.05). Crude protein values for SB ranged from 38.57% in raw sample to 47.46% (S2/30; sample soaked and autoclaved at 121 °C, 17 psi for 30 min), while that of MB ranged from 34.80% (raw) to 36.50% (S2/30; sample soaked and autoclaved at 121 °C, 17 psi for 30 min). The CP was found to be higher in the treated samples of both beans than in the raw samples, with soaked and autoclaved (S2/30; sample soaked and autoclaved at 121 °C, 17 psi for 30 min) having the highest CP value. The crude protein content of MB subjected to soaking (S), cooking for 20 min, steam autoclaving at 110 °C, 7 psi for 5 min (A1/5) and 30 min (A1/30), and S1/15, S2/5, S2/15, and S2/30 treatments did not differ, and these substrates had the highest CP values.

### 3.2. Amino Acids Composition of the Soybean and Marama Bean

The amino acids concentrations of raw and treated MB and SB samples are presented in Table 4. There were significant (*p* < 0.05) differences by substrate type, treatments, and interaction between substrate and treatments for all AAs, except for serine content, which showed no significant substrate effect. For raw samples, SB had higher values (*p* < 0.05) of lysine, threonine, leucine, glutamic acid, and alanine than MB, while valine, arginine, aspartic acid, phenylalanine, glycine, histidine, tyrosine, and proline were higher (*p* < 0.05) in MB than SB. Raw SB and MB had similar (*p* > 0.05) isoleucine content. In comparison to raw substrates, soaking for 24 h significantly reduced all the AAs (except aspartic acid, glutamic acid, and serine in SB; isoleucine and valine in MB) concentrations in both SB and MB. In addition, soaking SB for 24 h resulted in the lowest values for alanine (1.43%), arginine (2.43%), glycine (1.45%), histidine (0.87%), isoleucine (1.34%), leucine (2.54%), lysine (2.14%), phenylalanine (1.67%), threonine (1.30%), tyrosine (1.21%), and valine (1.49%). The loss in AAs in SB ranged from 2 to 15%, and valine was mostly impacted by 24 h soaking (1.75 vs. 1.49%). Soaking also resulted in highest loss in isoleucine (13%), histidine (7%), arginine (6%), leucine (6%), phenylalanine (6%), tyrosine (6%), and alanine (5%). Physical treatments yielded aspartic acid, glutamic acid, serine, and proline values that were equal to or greater than those in raw SB. Meanwhile, soaking MB for 24 h (S) resulted in the lowest values for alanine (1.2%), aspartic acid (3.78%), leucine (2.27%), lysine (1.94%), serine (2.04%), and threonine (1.14%). Presoaked steam-autoclaving at 121 °C and 17 psi for 30 min (S2/30) resulted in lowest values for arginine (2.35%), glutamic acid (4.0%), glycine (2.35%), isoleucine (0.98%), and tyrosine (3.61%) in MB. The adverse effect of thermal treatment on AAs was greater in MB than SB. The loss of AAs content of MB was as high as 36% (1.54% in raw reduced to 0.98%), as found with isoleucine in MB treated with presoaked steam autoclaving (S2/30), with the lowest being a 5% loss with lysine and leucine. Other essential amino acids in MB, including phenylalanine, histidine, and threonine, were lost up to 11% (caused by steam autoclaving at 110 °C and 7 psi for 5 min; A1/5), 8% (due to presoaked and autoclaved at 121 °C and 17 psi for 5 and 30 min, as well as steam autoclaving 110 °C and 7 psi for 15 min), and 7% (soaking, cooking for 20 min and presoaked and autoclaved at 110 °C and 7 psi for 5 min), respectively.

Some physical treatments, however, yielded an increase in AAs values of MB. Steam-autoclaving at 110 °C and 7 psi for 30 min (A1/30) yielded the highest values for alanine (1.56%), glutamic acid (5.20%), histidine (1.26%), leucine (2.93%), lysine (2.57%), phenylalanine (2.31%), proline (3.93%), serine (2.69%), threonine (1.49%), and valine (2.28%). The highest values for arginine (2.88%) and aspartic acid (4.63%) were recorded in MB samples cooked for 10 min (C10). Steam autoclaving at 121 °C (A2/30) resulted in the highest values for glycine (2.81%) and isoleucine (1.87%). Tyrosine was reduced in the MB samples by all the physical treatments, with the highest value of 5.10% being observed in the raw sample.

For SB, presoaked steam-autoclaving at 121 °C and 17 psi for 15 min (S2/15) resulted in highest values for glycine (2.45%), leucine (3.79%), phenylalanine (2.81%), serine (2.74%), isoleucine (1.96%), and threonine (1.87%). However, presoaked steam-autoclaving at 110 °C and 7 psi for 15 min (S1/15) and presoaked steam-autoclaving at 121 °C and 17 psi for 15 min (S2/15) yielded the highest values for isoleucine and threonine in SB that did not differ (*p* > 0.05). Presoaked steam-autoclaving of SB at 110 °C and 7 psi for 15 min (S1/15) resulted in the highest values for aspartic acid (5.41%), glutamic acid (7.11%), lysine (2.88%), and proline (2.63). The highest values for alanine (2.05%), arginine (3.55%), tyrosine (2.06%), and valine (2.37%) were found in the SB samples treated with steam-autoclaving at 110 °C and 7 psi for 15 min (A1/15). Histidine value was maximized (1.37%) by steam-autoclaving at 121 °C and 17 psi for 30 min.

### 3.3. Assessment of Trypsin Inhibitor Activity in Soybean and Marama Bean

The TIA in raw and treated MB and SB samples is shown in Table 5. There were significant substrate, treatment, and substrate × treatment interaction effects on TIA. For SB, cooking (C30), steam-autoclaving (A1/30, A2/5, A2/15, and A2/30), and pre-soaked autoclaving (S1/30, S2/5, S2/15, S2/30) resulted in a similar TUI (*p* > 0.05). Treated MB contained higher level of TIA, compared to treated SB. The largest reduction in TIA in SB (66.98 TUI/mg to 1.51 TUI/mg) was when the substrate was subjected to physical treatments A2/30. In MB, the TIA ranged from 245.25 TUI/mg (raw) to 2.66 TUI/mg (S2/30).

## 4. Discussion

### 4.1. Proximate Composition

The proximate composition of MB and soybean in the present study is similar to what has already been reported in the literature [21,22]. The decrease in DM observed in soaked and thermal-treated samples, compared to the raw samples, may be attributed to sample loss when soaking, cooking, or autoclaving water [23]. The ash value for SB was found to be higher than that in MB, which is in agreement with previous findings [3]. Meanwhile, the ash content of 2.88% (as is) recorded for MB was comparable to the 3% (as is) in previous findings [21]. However, the ash content was lower in the treated samples of both SB and MB, especially with samples that were soaked or steamed before autoclaving. This may point to a loss of nutrients in water during the physical treatments. The crude fat in MB was about three times higher than in SB, and the value compares favorably with the 40% crude fat content of MB that was reported earlier [21]. The NDF and ADF were found to be lower in MB than in soybean. Marama bean compares well with SB, in terms of the CP (34.80 vs. 38.57% DM basis) in raw samples, as recorded in this study, and this corroborates the earlier findings [3,6]. Crude protein was found to be higher in treated samples of both beans than the raw samples. This result was in line with previous findings [24] of an increase in the CP content of raw kidney beans when subjected to boiling. This suggests a release of nutrients that are bound to antinutritional factor by the thermal treatment methods in this study. For instance, phytate is naturally occurring in MB as a phytate-protein complex that can partially block the availability of the MB protein [21], and thermal treatment has been reported to reduce the phytic acid in cowpea [14].

### 4.2. Amino Acids

In this present study, the difference in AA levels of SB and MB, especially with higher aromatic AAs (tyrosine and phenylalanine) and proline in MB, despite the similarity in the crude protein contents of these two legumes, have been previously reported [13,20]. Similarly, findings from this study show that tyrosine, glutamic acid, aspartic acid, and proline are the major AAs in MB, which aligns with the earlier reports that assessed purified marama protein extract [25] and MB flour [20].

The increase in the concentration of some AAs in substrates subjected to physical treatments may be associated with the increase in CP values. On the other hand, the loss in concentrations of AA recorded with certain treatments (especially soaking and presoaked steam-autoclaving at 121 °C and 17 psi for 30 min) could have been caused by the partial loss of essential and non-essential AAs, with other nitrogenous compounds formed as protein was chemically degraded into water soluble AAs, due to prolonged soaking, thermal treatment, and/or pressure [26]. The recorded reduction in AAs content agrees with previous report of significant effects on the AAs profile of SB samples autoclaved for different durations, i.e., 0, 5, 10, 15, and 20 min [27].

### 4.3. Trypsin Inhibitor Activity

The TIA of raw MB was almost four times higher than that found in soybean. The value of 245.25 TUI/mg MB was close to the 250.8 TUI/mg earlier reported [20]. However, the TIA in raw SB of 66.98 TUI/mg reported in this study was higher than 57.6 TUI/mg previously recorded [20]. The difference observed in the TIA of the soybean may be due to differences in their variety and/or origin [28]. Further, the cooked (100 °C) and steam autoclaved (110 °C) soybean TIA values recorded in this study agree with those reported in a study by [29]. Soaking has been proven to be less effective in reducing TIA, with a 16% reduction in TIA in cowpea after 16 h of soaking [14], and this study was not different for SB soaked for 24 h, which resulted in 19% reduction in TIA (66.98 vs. 54.04 TUI/mg). The TIA was found to decline progressively by up to 96% (66.98 vs. 2.49 TUI/mg) and 96% (245.23 vs. 9.12 TUI/mg) as cooking time increased to 30 min (C30) for both soybean and MB, respectively. The lowest values for TIA for SB and MB were achieved with steam-autoclaving at 121 °C and 17 psi for 30 min, with (S2/30) or without presoaking (A2/30). Higher TIA reduction rate with pressure cooking (1 kg/cm^2^; 14.2 psi for 20 min) than cooking at 100 °C for 45 min has been reported earlier [14]. This implies that, despite the TIA of raw MB being about four times higher than that found in raw SB, the application of thermal treatment was able to reduce it by 99% (245.23 vs. 2.66 TUI/mg) and 98% (66.98 vs. 1.51 TUI/mg) TIA in MB and SB, respectively. This shows that the TIs in both SB and MB are heat-labile, and heating SB and MB at the temperature of 121 °C and 17 psi for 30 min was effective in modifying the conformation of the inhibitors, hence permanently inactivating them. Thermal processing of soybean for animal feed has been shown to increase body weight gain and feed efficiency in animals [17]. Specifically, the dietary inclusion of autoclaved (121 °C for 20 and 30 min) soybean meal improved growth performance and feed efficiency in broiler chickens, when compared with those fed raw SBM [17]. In this study, this treatment (autoclaving at 121 °C, 17 psi for 30 min) resulted in the lowest TIA values, which could explain the previously reported gains in the nutritive value of autoclaved MB upon incorporation in animal feed.

## 5. Conclusions

Raw marama bean contains similar crude protein content as raw soybean, but the AAs concentration and trypsin inhibitor activity vary greatly between these two substrates. The concentrations of alanine, glutamic acid, leucine, lysine, and threonine are higher in soybean than in marama bean, while marama bean crude protein contain more arginine, glycine, proline, and tyrosine than soybean. The trypsin inhibitor concentrations are about four times higher in marama bean than soybean. Physical treatments, especially prolonged (24 h) soaking, prolonged steam-autoclaving, and/or high pressure, reduced the ash and AAs contents in both beans. Conventional cooking and steam-autoclaving are effective in reducing or eradicating TIA in both seeds, thus enhancing the nutritional value. However, they may result in the loss of essential nutrients, especially essential AAs.

## Figures and Tables

**Table 1 molecules-27-04451-t001:** Description of samples.

Sample (SB/MB) ^1^	Description
R	Raw
S	Soaked for 24 h
C10	Cooking for 10 min
C20	Cooking for 20 min
C30	Cooking for 30 min
A1/5	Steam autoclaving at 110 °C and 7 psi for 5 min
A1/15	Steam autoclaving at 110 °C and 7 psi for 15 min
A1/30	Steam autoclaving at 110 °C and 7 psi for 30 min
A2/5	Steam autoclaving at 121 °C and 17 psi for 5 min
A2/15	Steam autoclaving at 121 °C and 17 psi for 15 min
A2/30	Steam autoclaving at 121 °C and 17 psi for 30 min
S1/5	Presoaked for 24 h and steam autoclaving at 110 °C and 7 psi for 5 min
S1/15	Presoaked for 24 h and steam autoclaving at 110 °C and 7 psi for 15 min
S1/30	Presoaked for 24 h and steam autoclaving at 110 °C and 7 psi for 30 min
S2/5	Presoaked for 24 h and steam autoclaving at 121 °C and 17 psi for 5 min
S2/15	Presoaked for 24 h and steam autoclaving at 121 °C and 17 psi for 15 min
S2/30	Presoaked for 24 h and steam autoclaving at 121 °C and 17 psi for 30 min

^1^ Sample: SB/MB = soybean/marama bean.

**Table 2 molecules-27-04451-t002:** Proximate composition (% as is, except for gross energy) of raw soybean and marama bean samples.

	Dry Matter	Ash	Crude Protein	Crude Fat	NDF	ADF	Gross Energy (Joule/g)
Soybean	90.22 (0.05)	4.57 (0.55)	34.79 (0.21)	17.22 (1.13)	14.73 (5.03)	8.34 (0.31)	22,800.33 (3.62)
Marama bean	94.03 (0.11)	2.88 (1.63)	32.72 (0.37)	38.67 (1.57)	7.51 (4.02)	3.43(0.36)	27,982.00 (3.12)

Values in parenthesis indicate covariance (CV); NDF = neutral detergent fiber; ADF = acid detergent fiber.

**Table 3 molecules-27-04451-t003:** Chemical composition (% DM, except for moisture) of raw and treated soybean and marama bean samples.

		Treaments ^1^																	
Parameters	Sub ^3^	R	S	C10	C20	C30	A1/5	A1/15	A1/30	A2/5	A2/15	A2/30	S1/5	S1/15	S1/30	S2/5	S2/15	S2/30	SEM ^2^
Moisture	SB	9.78 ^aA^	5.50 ^klA^	7.16 ^bcdA^	7.66 ^bA^	7.68 ^bA^	6.43 ^e–hA^	7.35 ^bcA^	6.95 ^cdeA^	6.03 ^h–kA^	6.28 ^f–iA^	4.54 ^mnoA^	6.02 ^h–kA^	6.80 ^c–fA^	6.62 ^d–gA^	6.22 ^f–jA^	6.17 ^g–jA^	5.12 ^mlA^	2.04
	MB	5.97 ^h–kB^	4.60 ^mnB^	5.65 ^jklB^	5.90 ^h–kB^	5.78 ^ijkB^	4.54 ^noB^	4.52 ^noB^	3.70 ^pB^	4.50 ^noB^	4.49 ^noB^	3.87 ^pB^	4.47 ^noB^	4.56 ^mnoB^	3.60 ^pB^	4.51 ^noB^	3.99 ^opB^	2.68 ^qB^	
DM ^4^	SB	90.22 ^cB^	87.50 ^hB^	89.51 ^dB^	88.33 ^fgB^	88.24 ^fgB^	88.46 ^fB^	87.03 ^ijkB^	86.05 ^lmB^	87.26 ^hijB^	86.78 ^kB^	86.10 ^lmB^	86.98 ^jkB^	86.20 ^lB^	85.38 ^nB^	86.78 ^kB^	85.84 ^mB^	85.24 ^nB^	0.16
	MB	94.03 ^aA^	89.40 ^deA^	91.30 ^bA^	90.13 ^cA^	89.08 ^eA^	90.36 ^cA^	90.23 ^eA^	89.48 ^dA^	89.50 ^dA^	89.51 ^dA^	89.12 ^eA^	89.10 ^eA^	88.30 ^fgA^	88.07 ^gA^	87.49 ^hA^	87.51 ^hA^	87.30 ^hiA^	
Ash	SB	5.07 ^aA^	4.44 ^bcA^	4.66 ^bA^	4.40 ^cA^	4.26 ^cA^	3.68 ^dA^	3.49 ^deA^	3.51 ^deA^	3.69 ^dA^	3.58 ^deA^	3.50 ^deA^	3.16 ^gA^	3.41 ^efA^	3.61 ^deA^	3.07 ^gA^	3.47 ^deA^	3.21 ^fgA^	0.85
	MB	3.06 ^gB^	2.53 ^hB^	2.40 ^hB^	2.32 ^hB^	2.48 ^hB^	2.07 ^jB^	1.92 ^jB^	2.09 ^jB^	1.95 ^jB^	2.09 ^jB^	1.96 ^jB^	1.89 ^jB^	1.93 ^jB^	1.98 ^jB^	2.09 ^ijB^	2.05 ^jB^	2.06 ^jB^	
CP ^5^	SB	38.57 ^jA^	41.70 ^fgA^	39.40 ^ijA^	40.35 ^hiA^	41.12 ^hgA^	42.17 ^fA^	45.33 ^cdeA^	44.62 ^eA^	42.22 ^fA^	44.41 ^eA^	44.39 ^eA^	44.80 ^eA^	47.25 ^abA^	46.31 ^bcA^	45.11 ^deA^	46.05 ^cdA^	47.46 ^aA^	3.63
	MB	34.80 ^nB^	36.87 ^klmB^	35.78 ^mnB^	36.46 ^klmB^	35.95 ^lmB^	36.42 ^klmB^	36.84 ^klB^	36.62 ^klmB^	36.89 ^klB^	35.91 ^lmB^	37.17 ^kB^	37.02 ^kB^	36.60 ^klmB^	36.88 ^klB^	36.36 ^klmB^	36.60 ^klmB^	36.50 ^klmB^	

^1^ Treatments: R = raw soybean/marama bean; S = soybean/marama bean sample soaked for 24 h; C10 = soybean/marama bean cooked for 10 min; C20 = soybean/marama bean cooked for 20 min; C30 = soybean/marama bean cooked for 30 min; A1/5 = steam soybean/marama bean sample autoclaved at 110 °C, 7 psi for 5 min; A1/15 = steam soybean/marama bean sample autoclaved at 110 °C, 7 psi for 15 min; A1/30 = steam soybean/marama bean sample autoclaved at 110 °C, 7 psi for 30 min; A2/5 = steam soybean/marama bean sample autoclaved at 121 °C, 17 psi for 5 min; A2/15 = steam soybean/marama bean sample autoclaved at 121 °C, 17 psi for 15 min; A2/30 = steam soybean/marama bean sample autoclaved at 121 °C, 17 psi for 30 min; S1/5 = soybean/marama bean sample soaked and autoclaved at 110 °C, 7 psi for 5 min; S1/15 = soybean/marama bean sample soaked and autoclaved at 110 °C, 7 psi for 15 min; S1/30 = soybean/marama bean sample soaked and autoclaved at 110 °C, 7 psi for 30 min; S2/5 = soybean/marama bean sample soaked and autoclaved at 121 °C, 17 psi for 5 min; S2/15 = soybean/marama bean sample soaked and autoclaved at 121 °C, 17 psi for 15 min; S2/30 = soybean/marama bean sample soaked and autoclaved at 121 °C, 17 psi for 30 min. ^2^ SEM = standard error of mean. ^3^ Sub = substrate (SB, soybean; MB, marama bean). ^4^ DM = dry matter. ^5^ CP = crude protein. ^a–q^ Means within each row with no common superscripts are significantly different (*p* < 0.05). ^A,B^ Means within each column with no common superscripts are significantly different (*p* < 0.05).

**Table 4 molecules-27-04451-t004:** Amino acids composition (%, dry matter basis) of raw and treated marama and soybean samples.

Parameter	Sp ^3^	R	S	C10	C20	C30	A1/5	A1/15	A1/30	A2/5	A2/15	A2/30	S1/5	S1/15	S1/30	S2/5	S2/15	S2/30	SEM ^2^
Alanine	SB	1.49 ^f–kA^	1.43 ^h–lA^	1.69 ^deA^	1.55 ^e–jA^	1.82 ^bcdA^	1.59 ^e–hA^	2.05 ^aA^	1.93 ^abcA^	1.71 ^deA^	1.93 ^abcA^	1.66 ^defA^	1.81 ^cdA^	2.01 ^abA^	1.64 ^d–gA^	1.91 ^abcA^	1.95 ^abcA^	1.81 ^cdA^	0.07
	MB	1.31 ^k–nB^	1.20 ^nB^	1.45 ^g–lB^	1.24 ^mnB^	1.32 ^k–nB^	1.31 ^k–nB^	1.28 ^lmnB^	1.56 ^e–iB^	1.35 ^k–nB^	1.27 ^lmnB^	1.45 ^g–lB^	1.23 ^nB^	1.25 ^mnB^	1.39 ^i–nB^	1.28 ^lmnB^	1.38 ^i–nB^	1.36 ^j–nB^	
Arginine	SB	2.48 ^klmB^	2.34 ^mB^	2.94 ^d–gA^	2.85 ^e–kA^	3.03 ^c–fA^	3.19 ^a–eA^	3.55 ^aA^	3.19 ^a–eA^	2.86 ^e–jA^	3.16 ^b–eA^	3.31 ^abcA^	3.14 ^b–eA^	3.35 ^abcA^	2.66 ^g–mA^	3.48 ^abA^	3.31 ^abcA^	2.92 ^e–hA^	0.13
	MB	2.71 ^f–mA^	2.51 ^j–mA^	2.88 ^e–iB^	2.48 ^klmB^	2.56 ^h–mB^	2.51 ^j–mB^	2.52 ^i–mB^	3.14 ^b–eB^	2.72 ^f–lB^	2.49 ^klmB^	2.93 ^e–hB^	2.50 ^j–mB^	2.60 ^g–mB^	2.62 ^g–mA^	2.42 ^lmB^	2.65 ^g–mB^	2.35 ^mB^	
Aspartic acid	SB	4.00 ^i–lB^	4.11 ^h–lA^	4.45 ^e–kB^	4.25 ^f–lA^	4.79 ^b–fA^	4.78 ^b–fA^	5.08 ^abcA^	5.11 ^abA^	4.51 ^c–jA^	5.01 ^a–dA^	3.90 ^klB^	4.76 ^b–gA^	5.41 ^aA^	4.41 ^e–kA^	4.96 ^a–eA^	5.04 ^abcA^	4.64 ^b–hA^	0.21
	MB	4.08 ^h–lA^	3.78 ^lB^	4.63 ^b–hA^	3.81 ^lB^	4.12 ^h–lB^	4.04 ^i–lB^	3.99 ^i–lB^	4.95 ^a–eB^	4.22 ^f–lB^	3.94 ^jklB^	4.54 ^b–iA^	3.82 ^lB^	3.88 ^klB^	4.20 ^g–lB^	3.89 ^klB^	4.17 ^h–lB^	4.19 ^g–lB^	
Glutamic acid	SB	5.24 ^f–iA^	5.24 ^f–iA^	6.00 ^b–fA^	5.75 ^e–hA^	6.30 ^b–eA^	5.30 ^f–iA^	6.72 ^abA^	6.63 ^abcA^	5.87 ^c–gA^	6.50 ^a–eA^	5.31 ^f–iA^	6.28 ^b–eA^	7.11 ^aA^	5.76 ^e–hA^	6.56 ^a–dA^	6.47 ^a–eA^	5.83 ^d–gA^	0.28
	MB	4.98 ^hijB^	4.57 ^ijkB^	5.25 ^f–iB^	4.30 ^jkB^	4.55 ^ijkB^	4.18 ^kB^	4.20 ^kB^	5.20 ^ghiB^	4.41 ^jkB^	3.93 ^kB^	5.04 ^hijB^	3.94 ^kB^	4.10 ^kB^	4.19 ^kB^	3.95 ^kB^	4.10 ^kB^	4.00 ^kB^	
Glycine	SB	1.55 ^opB^	1.45 ^pB^	1.89 ^k–nB^	1.81 ^mnB^	2.00 ^j–lB^	2.08 ^jkB^	2.35 ^ghiB^	2.06 ^jkB^	1.83 ^lmnB^	2.03 ^j–lB^	2.43 ^fghB^	2.05 ^j–klB^	2.13 ^ijB^	1.73 ^noB^	2.19 ^hijB^	2.45 ^efgB^	1.88 ^k–nB^	0.08
	MB	2.70 ^bcdA^	2.44 ^efgA^	2.73 ^bcA^	2.51 ^c–gA^	2.46 ^efgA^	2.44 ^efgA^	2.46 ^efgA^	3.04 ^aA^	2.67 ^b–eA^	2.47 ^d–gA^	2.81 ^abA^	2.44 ^efgA^	2.54 ^c–gA^	2.63 ^b–fA^	2.47 ^d–gA^	2.56 ^c–gA^	2.35 ^ghiA^	
Histidine	SB	0.94 ^lmB^	0.87 ^mB^	1.10 ^e–lB^	1.05 ^g–lA^	1.15 ^d–kA^	1.32 ^a–dA^	1.26 ^a–eA^	1.18 ^c–hA^	1.05 ^g–lA^	1.16 ^d–jA^	1.37 ^abA^	1.20 ^b–gA^	1.24 ^a–fA^	1.01 ^h–mB^	1.39 ^aA^	1.35 ^abcA^	1.08 ^f–lA^	0.06
	MB	1.07 ^f–lA^	1.00 ^i–mA^	1.12 ^e–kA^	1.01 ^h–mB^	1.01 ^h–mB^	1.00 ^j–mB^	0.98 ^klmB^	1.26 ^a–eB^	1.10 ^e–lB^	1.00 ^j–mB^	1.18 ^c–hB^	1.00 ^i–mB^	1.04 ^g–mB^	1.06 ^g–lA^	0.98 ^klmB^	1.07 ^f–lB^	0.98 ^klmB^	
Isoleucine	SB	1.54 ^ijkA^	1.34 ^kB^	1.72 ^e–iA^	1.46 ^jkB^	1.87 ^b–eA^	1.83 ^b–fA^	2.15 ^aA^	1.89 ^b–eB^	1.60 ^g–iB^	1.96 ^a–dA^	1.77 ^d–hB^	1.84 ^b–fA^	2.03 ^abA^	1.64 ^e–iB^	1.95 ^a–dA^	1.96 ^a–dA^	1.86 ^b–eA^	0.08
	MB	1.54 ^ijkA^	1.55 ^ijkA^	1.70 ^e–iA^	1.59 ^g–iA^	1.64 ^f–iB^	1.70 ^e–iB^	1.61 ^g–iB^	2.00 ^abcA^	1.76 ^d–hA^	1.52 ^ijkB^	1.87 ^b–eA^	1.60 ^g–iB^	1.58 ^hijB^	1.71 ^e–iA^	1.58 ^hijB^	1.80 ^c–gB^	0.98 ^lB^	
Leucine	SB	2.71 ^jklA^	2.54 ^l–oA^	3.05 ^ghiA^	2.82 ^ijkA^	3.27 ^efgA^	3.31 ^defA^	3.76 ^abA^	3.50 ^cdA^	3.11 ^fghA^	3.53 ^bcdA^	3.41 ^deA^	3.41 ^deA^	3.68 ^abcA^	3.00 ^hiA^	3.65 ^abcA^	3.79 ^aA^	3.37 ^deA^	0.08
	MB	2.41 ^m–pB^	2.27 ^pB^	2.63 ^klmB^	2.31 ^opB^	2.40 ^m–pB^	2.37 ^nopB^	2.38 ^nopB^	2.93 ^hijB^	2.56 ^lmnB^	2.38 ^nopB^	2.71 ^jklB^	2.33 ^nopB^	2.39 ^nopB^	2.48 ^l–pB^	2.39 ^m–pB^	2.55 ^lmnB^	2.56 ^lmnB^	
Lysine	SB	2.18 ^g–oA^	2.14 ^i–oA^	2.44 ^c–gA^	2.22 ^f–nA^	2.60 ^bcdA^	2.56 ^b–eA^	2.60 ^bcdA^	2.66 ^abcA^	2.33 ^e–jA^	2.67 ^abcA^	2.23 ^f–nB^	2.55 ^b–eA^	2.88 ^aA^	2.32 ^e–jA^	2.48 ^b–fA^	2.72 ^abA^	2.39 ^d–i^	0.09
	MB	2.04 ^l–oB^	1.94 ^oB^	2.42 ^c–gA^	1.94 ^oB^	2.19 ^g–oB^	2.20 ^g–oB^	2.11 ^j–oB^	2.57 ^b–eB^	2.17 ^h–oB^	1.99 ^noB^	2.36 ^d–jA^	2.00 ^mnoB^	2.04 ^l–oB^	2.21 ^g–nB^	2.09 ^k–oB^	2.26 ^f–lB^	2.25 ^f–m^	
Phenylalanine	SB	1.77 ^jkB^	1.67 ^kB^	2.15 ^d–jA^	2.63 ^abcA^	2.27 ^c–iA^	2.59 ^abcA^	2.82 ^aA^	2.38 ^b–eA^	2.15 ^d–jA^	2.41 ^a–eA^	2.65 ^abcA^	2.37 ^b–fA^	2.48 ^a–dA^	2.03 ^e–kA^	2.75 ^abA^	2.81 ^aA^	2.28 ^c–hA^	0.14
	MB	2.02 ^e–kA^	1.86 ^ijkA^	2.02 ^e–kB^	1.89 ^h–kB^	1.86 ^ijkB^	1.81 ^jkB^	1.85 ^jkB^	2.31 ^c–gB^	2.05 ^e–kB^	1.95 ^g–kB^	2.14 ^d–jB^	1.86 ^ijkB^	1.93 ^g–kB^	1.92 ^g–kB^	1.86 ^ijkB^	1.97 ^f–kB^	1.93 ^g–kB^	
Proline	SB	1.90 ^jB^	1.81 ^jB^	2.18 ^hiB^	1.97 ^ijB^	2.40 ^fghB^	2.58 ^fB^	1.98 ^ijB^	2.52 ^fgB^	2.18 ^hiB^	2.48 ^fgB^	2.48 ^fgB^	2.48 ^fgB^	2.63 ^fB^	2.18 ^hiB^	2.52 ^fgB^	3.14 ^eB^	2.29 ^ghB^	0.09
	MB	3.46 ^bcdA^	3.16 ^eA^	3.52 ^bcA^	3.24 ^deA^	3.17 ^eA^	3.11 ^eA^	3.26 ^cdeA^	3.93 ^aA^	3.46 ^bcdA^	3.22 ^deA^	3.53 ^bA^	3.18 ^eA^	3.23 ^deA^	3.31 ^b–eA^	3.34 ^b–eA^	3.26 ^cdeA^	3.20 ^eA^	
Serine	SB	1.88 ^m^	1.87 ^m^	2.16 ^i–l^	2.04 ^lm^	2.25 ^e–k^	2.37 ^c-i^	2.5 ^a–d^	2.44 ^c–g^	2.25 ^f-l^	2.40 ^c-h^	2.38 ^c–i^	2.32 ^c–j^	2.51 ^a–d^	2.10 ^j–m^	2.69 ^ab^	2.74 ^a^	2.23 ^g–l^	0.08
	MB	2.26 ^e–l^	2.04 ^lm^	2.54 ^abc^	2.06 ^klm^	2.22 ^g–l^	2.18 ^h–l^	2.13 ^jkl^	2.69 ^b^	2.30 ^d–j^	2.22 ^g–l^	2.48 ^b–f^	2.09 ^j–m^	2.25 ^f–l^	2.49 ^b–e^	2.17 ^h–l^	2.39 ^c–i^	2.23 ^f–l^	
Threonine	SB	1.32 ^h–kA^	1.30 ^i–lA^	1.58 ^defA^	1.51 ^efgA^	1.64 ^b–fA^	1.57 ^defA^	1.79 ^abcA^	1.72 ^a–dA^	1.58 ^defA^	1.73 ^a–dA^	1.66 ^b–eA^	1.62 ^c–fA^	1.80 ^abA^	1.48 ^fghA^	1.87 ^aA^	1.87 ^a^	1.59 ^defA^	0.06
	MB	1.22 ^i–lB^	1.14 ^lB^	1.38 ^ghiB^	1.14 ^lB^	1.23 ^i–lB^	1.20 ^klB^	1.19 ^klB^	1.49 ^fgB^	1.28 ^i–lB^	1.21 ^jklB^	1.37 ^g–jB^	1.15 ^lB^	1.21 ^jklB^	1.26 ^i–lB^	1.17 ^klB^	1.29 ^i–lB^	1.25 ^i–lB^	
Tyrosine	SB	1.29 ^lmB^	1.21 ^mB^	1.57 ^j–mB^	1.45 ^klmB^	1.65 ^i–lB^	1.69 ^i–lB^	2.06 ^hiB^	1.74 ^ijkB^	1.56 ^j–mB^	1.73 ^ijkB^	2.44 ^hB^	1.74 ^ijkB^	1.78 ^ijkB^	1.48 ^klmB^	1.99 ^iB^	1.95 ^ijB^	1.66 ^i–lB^	0.15
	MB	5.10 ^aA^	4.52 ^bcA^	4.70 ^abA^	4.25 ^cdA^	4.07 ^defA^	3.61 ^gA^	3.80 ^efgA^	4.73 ^abA^	4.12 ^cdeA^	3.65 ^gA^	4.79 ^abA^	3.64 ^gA^	3.91 ^d–gA^	3.69 ^fgA^	3.65 ^gA^	3.59 ^gA^	3.53 ^gA^	
Valine	SB	1.75 ^jkA^	1.49 ^lB^	1.91 ^e–jB^	1.65 ^klB^	2.07 ^b–fA^	2.02 ^c–iA^	2.37 ^aA^	2.07 ^b–fB^	1.76 ^jkB^	2.17 ^a–d^	1.95 ^d–jB^	2.01 ^c–iA^	2.24 ^abcA^	1.85 ^f–jB^	1.76 ^jkB^	2.23 ^abcA^	2.05 ^c–hA^	0.08
	MB	1.77 ^jkA^	1.78 ^jkA^	1.95 ^d–jA^	1.83 ^g–kA^	1.88 ^f–iB^	1.95 ^d–jB^	1.84 ^g–kB^	2.28 ^abA^	2.01 ^c–iA^	1.74 ^kB^	2.13 ^b–eA^	1.82 ^h–kB^	1.80 ^ijkB^	1.97 ^d–jA^	1.80 ^ijkA^	2.05 ^b–gB^	2.04 ^c–hA^	

^2^ SEM = standard error of mean. ^3^ Sp = substrates (SB, soybean; MB, marama bean). ^a–p^ Means within each row with no common superscripts are significantly different (*p* < 0.05). ^A,B^ Means within each column with no common superscripts are significantly different (*p* < 0.05).

**Table 5 molecules-27-04451-t005:** Trypsin inhibitor activity (TUI /mg) of raw and treated soybean and marama bean samples.

		Treaments ^1^																	
Parameter	Sub ^4^	R	S	C10	C20	C30	A1/5	A1/15	A1/30	A2/5	A2/15	A2/30	S1/5	S1/15	S1/30	S2/5	S2/15	S2/30	SEM ^2^
TUI ^3^	SB	66.98 ^cB^	54.04 ^dB^	11.52 ^gB^	5.21 ^hijB^	2.49 ^jB^	3.69 ^hijB^	3.09 ^hijB^	2.49 ^jB^	1.89 ^JB^	1.89 ^jB^	1.51 ^jB^	3.38 ^hijB^	2.69 ^ijB^	2.19 ^jB^	1.91 ^jB^	2.12 ^jB^	1.60 ^jB^	2.22
	MB	245.25 ^bA^	271.98 ^aA^	47.08 ^eA^	18.22 ^fA^	9.12 ^ghA^	8.85 ^ghiA^	7.32 ^g–jA^	5.40 ^g–jA^	6.56 ^g–iA^	4.23 ^hijA^	3.00 ^hijA^	7.73 ^g–jA^	5.76 ^g–jA^	5.05 ^hijA^	6.26 ^g–jA^	4.22 ^hijA^	2.66 ^ijA^	

^1^ Treatments: R = raw soybean/marama bean; S = soybean/marama bean sample soaked for 24 h; C10 = soybean/marama bean cooked for 10 min; C20 = soybean/marama bean cooked for 20 min; C30 = soybean/marama bean cooked for 30 min; A1/5 = steam soybean/marama bean sample autoclaved at 110 °C, 7 psi for 5 min; A1/15 = steam soybean/marama bean sample autoclaved at 110 °C, 7 psi for 15 min; A1/30 = steam soybean/marama bean sample autoclaved at 110 °C, 7 psi for 30 min; A2/5 = steam soybean/marama bean sample autoclaved at 121 °C, 17 psi for 5 min; A2/15 = steam soybean/marama bean sample autoclaved at 121 °C, 17 psi for 15 min; A2/30 = steam soybean/marama bean sample autoclaved at 121 °C, 17 psi for 30 min; S1/5 = soybean/marama bean sample soaked and autoclaved at 110 °C, 7 psi for 5 min; S1/15 = soybean/marama bean sample soaked and autoclaved at 110 °C, 7 psi for 15 min; S1/30 = soybean/marama bean sample soaked and autoclaved at 110 °C, 7 psi for 30 min; S2/5 = soybean/marama bean sample soaked and autoclaved at 121 °C, 17 psi for 5 min; S2/15 = soybean/marama bean sample soaked and autoclaved at 121 °C, 17 psi for 15 min; S2/30 = soybean/marama bean sample soaked and autoclaved at 121 °C, 17 psi for 30 min. ^2^ SEM = standard error of mean. ^3^ TUI= trypsin inhibitor unit. ^4^ Sub = substrates. ^a–j^ Means within each row with no common superscripts are significantly different (*p* < 0.05). ^A,B^ Means within each column with no common superscripts are significantly different (*p* < 0.05).

## Data Availability

Not applicable.

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
