# Peer review of "Physical Treatment Reduces Trypsin Inhibitor Activity and Modifies Chemical Composition of Marama Bean (Tylosema esculentum)"

_molecules, 2022, doi:10.3390/molecules27144451_

Round 1

Reviewer 1 Report

In materials and method;

- need harvesting year for soybean seed.

- treatment is too complicated (adjust treatment with not difference statistically).  

In discussion;

- compare TA of soybean obtained in this work to previous results.      

Reviewer 2 Report

Reviewer’s recommendation

The manuscript was well written and discussed about the innovative study of using thermal treatment for inactivation of trypsin inhibitor in marama bean. I recommend this manuscript for publication on Molecules journal after the authors address all comments below

1.    Did oven drying at 45-50°C (lines 102-104) completely remove water ? If not sure, authors should provide sample moisture content after oven drying.  

2.    The samples stated from line 107-116 should be described in a table that will help readers to follow easily.

3.    How did the authors quantify amino acids in lines 141-143 without using amino acid standards ?

4.    The greatest reduction of TIA was observed by heating MB and SB samples at the temperature of 121°C and 17 psi for 30 minutes. The authors should discuss further about the practical application of this treatment to prepare MB for animal feed.
